# An Analysis of Male and Female Managers’ Responses to Work Stress: Focused on the Case of South Korea

**DOI:** 10.3390/ijerph182111119

**Published:** 2021-10-22

**Authors:** Kyung-Ran Roh, Eun-Bee Kim

**Affiliations:** 1Department of Education, Education College, Sungshin Women’s University, Seoul 02844, Korea; nari123@sungshin.ac.kr; 2Department of Liberal Arts, Wesley Creative Convergence College, HyupSung University, Hwaseong 18330, Korea

**Keywords:** work stress, career development behaviors, capabilities, turnover intentions, managers

## Abstract

This study aims to determine how work the stress of female and male managers in Korean companies influences their capabilities and career development behavior and analyzes how it affects their turnover intention. According to the analysis results determined using the panel data from the Korean Women’s Development Institute, first, work stress experienced by managers increases their turnover intentions regardless of gender. Secondly, more work stress decreases the development behavior of male mangers but strengthens the career development behavior of female managers. Thirdly, greater work stress decreases manager capabilities but strengthens the capabilities of female managers. Fourthly, career development behavior and capabilities as a mediator variable act differently according to gender on the relationship between work stress and turnover intention. We discussed the results of this study while considering the peculiarities of Korea, which has strong male-centered corporate culture, and presented theoretical and practical implications of the results of this study.

## 1. Introduction

In a business environment where the speed of social change is very fast and where the future is highly uncertain, securing excellent human resources is very important for the survival of a company and as well as to gain a competitive advantage [1,2]. In particular, based on the fact that managerial capabilities have a great influence on the performance of the company [1,3], it can be said that human resource management for managers is the key to maintaining the competitiveness of the company. In other words, it is important to ensure that managers do not leave their organization by developing their careers and fully utilizing their capabilities to grow in their organization. From this point of view, it appears that work stress has a negative effect on career orientation [4], performance [5], and turnover intention [6], raising the need to pay attention to the stress that managers face in the workplace. This means the relationship between work stress, capabilities, career development behavior, and turnover intention should be comprehensively analyzed under recent career contexts where stress at work is increasing.

On the other hand, although it has been reported that many of managers feel huge stress at work [7,8], previous studies have highlighted the fact that managers are one of stressors for subordinates [9] or that managers are the people who are the people who are responsible for managing organizational stress [10]. There are relatively few studies on how managers should cope with stress at work with the perspective that they can also perceive stress as a human being and strive to survive in the organization. In today’s work environment, where stress is inevitable, in order for managers to successfully develop their careers within the organization while maintaining their mental health, it is necessary to analyze their responses to work stress in detail.

Therefore, the purpose of this study is to investigate the relationship between the work stress perceived by managers and career development behaviors, capabilities, and turnover intentions as the responses of mangers to stress in the workplace. In particular, this study intends to suggest implications that can contribute to enhancing diversity in management positions by comparatively analyzing differences according to gender in response to work stress under a generally male-dominated workplace culture [11,12]. To compare and analyze work stress response behaviors according to gender, data obtained from managers working in companies in Korea, where gender inequality in managerial positions is salient [13,14], was adopted. The characteristics of the Korean labor market in terms of gender gap are as follows: The economic participation rate of women (ages 15–64) increased by 7.1 percentage points from 46.3% in 1980 to 53.4% in 2021, although the Equal Employment Opportunity Act was enacted in 1987 to resolve the gender gap in the labor market. Compared to men’s economic participation rate (72.6%) in 2021, the women’s rate is 19.2% lower, and this gender gap is wider in the 30s and 40s age group, which is a key career formation period [15]. In addition, as of 2020, the wage of Korean women is 67.7% of that of men [15]. Of the total executives (32,005) of 2246 listed corporations, only 5.2% are women, and 63.7% of companies have no female executives [16]. Considering that the OECD average female executive ratio announced by The Economist in 2021 is 25.6%, it can be seen that women in the Korean labor market are under the influence of the extreme glass ceiling effect.

Meanwhile, under the Korean education system, where education is compulsory up to middle school, female high school enrollment rate was 96.6% (96.1% of the total), and the female college entrance rate was 76.0% (72.5% of the total), indicating that gender equality in educational opportunities was achieved to some extent [17]. Accordingly, the phenomenon of separation of majors by gender and separation of occupations has eased over the past 10 years. However, when it comes to the college majors of college graduates, males are biased towards engineering, and females are biased towards humanities and education. In addition, the phenomenon of gender occupational segregation, in which women are concentrated in traditional female occupations such as occupations related to health, social welfare, and education, remains [18].

## 2. Literature Review and Hypothesis Development

### 2.1. Theoretical Background

Work stress is defined as physical and emotional strain that employees perceive when they execute their tasks at their workplace [19,20]. Work-related stress has become a critical issue in the occupational health field because it presents problems at work such as absenteeism, turnover, poor performance, and so forth [21,22]. Specifically, the authors of the current work are interested in the relationship between work stress and turnover intention as key concerns for an organization. Turnover results in the loss of human resources and high costs and time spent on recruiting and placing newcomers to replace leavers immediately [23]; admittedly, turnover intention is a salient indicator of actual turnover behaviors [24].

Although work stress has been demonstrated as a robust predictor of the turnover intentions of health workers in previous studies [25,26,27], the relationship between work stress and turnover intention in other occupations or workers in other industries is still controversial [5,6,28,29]. Thus, the relationship between work stress and turnover intention need to be analyzed by utilizing data from surveys targeting employees working in all industries rather than a specific occupation/industry.

On the other hand, this research focuses on how managers cope with work stress in an inevitably stressful work environment [30] using the conservation of resources (COR) theory, which is useful to explain employee behaviors in the workplace. The COR theory explains employee desires to avoid central resource losses as well as to acquire resources for survival, which is essential to human behavior genetics [31]. COR theory posits that stress occurs (a) when central or key resources are threatened with loss, (b) when central or key resources are lost, or (c) when there is a failure to gain central or key resources following significant effort [31]. It is worth noting that this perspective aims to study human behavioral strategies in response to stress when it occurs for any reason. COR theory asserts that the motivation to avoid the loss of resources supersedes the motivation to acquire additional resources [31]. In other words, employees do not want to spend time on additional activities to maintain the level of energy they want to have when their energy resources are already starting to diminish [32,33]. In order to create continuous individual performance at work, it is necessary to examine how employees respond to stress in today’s business environment, where additional competency development is important beyond maintaining duties and tasks. In particular, the authors are interested in empirically verifying whether there are differences based on gender in terms of response to stress in a male-dominated corporate culture.

From this point of view, the basic question that the authors have is how managers behave when they perceive work stress in terms of human resource perspectives. According to COR logic, we presume that managers would have the desire to not improve their capabilities, namely not to undergo any career development behaviors, in order to protect against losses because of work stress rather than to leave. Previous studies have shown that work stress negatively affects employee capabilities [34,35] and career development behaviors [36,37]. Additionally, a negative empirical relationship has been examined between employee capabilities and turnover intentions [38,39]; likewise, career development behaviors diminished turnover intentions [40,41,42]. Although, the relationship between specific factors has been examined, the relationships between work stress, capabilities, career development behavior, and turnover intention are not clear synthetically.

Thus, the authors propose capabilities as a key resource to survive in a competitive workplace as well as career development behaviors as an investment resource to recover key resources based on COR theory, which will diminish the propensity that managers have to leave. To examine manager behaviors when coping with work stress, as stated based on above previous research, the following hypotheses are developed:

**Hypothesis** **1** **(H1):**
*Work stress will affect manager turnover intentions.*


**Hypothesis** **2** **(H2):**
*Work stress will affect manager career development behaviors.*


**Hypothesis** **3** **(H3):**
*Work stress will affect manager capabilities.*


**Hypothesis** **4** **(H4):**
*Career development behavior will play a moderating role in the relationship between work stress and turnover intentions.*


**Hypothesis** **5** **(H5):**
*Capabilities will play a moderating role in the relationship between work stress and turnover intention.*


### 2.2. The Differences between Female Managers and Male Managers

When it comes to work stress and gender, previous studies have focused on which gender is more vulnerable to work stress. Some studies found that female workers experience more stress than men [37,38], while others reported no difference between gender [39]. However, the difference in coping with stress at work between female and male workers has not been focused on. Although Haarr and Marash’ s work [40] showed that female police officers cope with stress by using escape and by keeping written records more often than men, this result is restricted to for police officers is not applicable to other occupations. Therefore, differences in reactions to work stress based on gender need to be explored through analyzing data surveying targeted employees working in all industries rather than a specific occupation/industry.

On the other hand, attempts to explain organizational inequality issues such as the glass ceiling, salary gaps, and so forth between male managers and female managers have been conducted since the 1970s. Several studies determined that the opportunity gap was due to education and training in terms of gender [43], while others argued that this phenomena was due to gender-role perspectives, such as job deployment [44], leadership style [45], behavior characteristics [46], career interruption [47,48], and organizational characteristics that were in favor of males [49] and social characteristics such as female confidence and aspirations for career success [50] as reasons. As one of the representative studies explaining career development behavior differences between female and male managers, Cannings and Montmarquette [51] analyzed the relationship between performance, career aspiration, and reward in the internal promotion process of companies and argued that females tended to rely more on formal promotion opportunities than males, which resulted in depriving females of opportunities. Males, on the other hand, were more inclined to use informal networks for promotion and to be promoted in ways such as showing their strong desire to attract more attention from their bosses, rather than depending on a performance system. Previous studies have focused more on exploring determinants of disproportionate status rather than their difference on career development behavior; nevertheless, female/male employees participated in career development activities differently according to Cannings and Montmarquette [52].

Thus, this research aims to investigate how female/male managers react differently in terms of career perspectives such as capabilities, career development behavior, and turnover intentions under a stressful workplace environment. The authors propose the behaviors used by employees to cope with work stress may be different based on gender, in contrast with the above previous studies. The five hypotheses that were developed in this study will be analyzed through a comparative perspective looking at gender.

## 3. Methods

### 3.1. Research Model

This study aimed to investigate the relationship between work stress perceived by managers in terms of career development behaviors, capabilities, and turnover intentions as responses to stress at the workplace. For this purpose, a research model was constructed, as shown in Figure 1.

### 3.2. Participants

The data used in this study are from government-funded research institutes under the Prime Minister’s Office that were established by the Korean Women’s Development Institute to study and develop policies in various fields related to women. The Women Manager Panel Survey is the only survey in Korea and abroad that can observe the employment and growth process of female managers in a company to understand the status of work, organizational culture, network, career development, leadership, and work–life balance. Since 2020, a panel of new female managers and male managers has been established, and a comparative analysis of the male and female managers is possible in order to produce the data necessary for policy establishment and evaluation on fostering female managers. The survey target is female managers at the level of assistant manager or higher who are employed in companies with 100 or more employees. Since this survey is focused on female career formation within the company, the level of assistant manager, which can be viewed as the initial stage of management, is included in the survey target. This survey is a national statistical survey approved by the National Statistical Office (approval number 154010). In the future, data from this survey will be the basis for policy establishment to resolve gender inequality and to expand female managers in the labor market [53].

This study was undertaken using data concerning 2279 managers working at companies in Korea that were obtained from the Korean Women’s Manager Panel Survey 2018(KWMPS 2018) conducted by Korean Women’s Development Institute. KWMPS 2018 was designed to establish data to view the status of work, organization culture, network, career development, leadership, and daily life balance by observing the employment and growth process of female managers in the companies. This panel survey was reviewed with Korean Women’s Development Institute IRB (Institutional Review Board) and was executed using the TAPI (Tablet Assisted Personal Interview) method. In addition, for the purpose of comparing female and male managers, an additional survey was conducted by extracting male managers as a sample from companies that extracted female managers in 2018. In other words, the male manager data used in this study should be understood as the results of fellow male managers in the same company, which were extracted for the purpose of comparison with female managers and do not represent all male managers in Korea [52].

### 3.3. Data Analysis and Research Procedures

The analysis was performed using the SPSS21.0 (IBM, New York, NY, USA) and PROCESS macro programs (Andrew F. Hayes, Calgary, AB, Canada). The analysis proceeded as follows: First, a frequency analysis was conducted on the independent variables, the dependent variables, and the mediator variables. Second, a confirmatory factor analysis was conducted to confirm the level of appropriateness of the measuring variables. Third, a correlation analysis was conducted to identify the direction of relevance and relationships among the variables. Fourth, a research model and an alternative model were constructed and compared to identify which model was appropriate to use for analysis. Fifth, a structural equation was used to conduct covariance structural analysis and to confirm whether there were influencing effects between the study variables as well as to identify the appropriateness and advantage of this research model. Sixth, a mediator analysis was undertaken for verification using the PROCESS macro. According to reference [50], the PROCESS macro is a useful analytic method for the verification of mediation and control effects in that it does not require going through separate procedures using regression analysis, such as those used by Baron and Kenny [53] or the Sobel test. The PROCESS macro uses bootstrapping to verify mediation effects where the basic number of samples has been set to at least 2000 and determines the resulting values in a single analysis.

## 4. Results

The verification results established in this study are as follows:

**Hypothesis** **1** **(H1):**
*Work stress affects manager turnover intentions.*


First, as shown in the results in Table 1, work stressed experienced by male managers has a positive influence on their turnover intentions. The value is 0.591, which is statistically significant (*p* < 0.001).

This shows that the when the workplace stress experienced by male managers increases, this stress causes an increase in their turnover intention.

Second, the work stress experienced by female managers has a positive influence on their turnover intention.

The value is 0.117, which is statistically significant (*p* < 0.001).

This suggests that when female managers experience increased work stress, it causes an increase in their turnover intention.

Therefore, the work stress experienced by managers increases their turnover intention regardless of gender. In addition, the value for the male managers was higher than that of the female managers.

**Hypothesis** **2** **(H2):**
*Work stress affects the career development behavior of managers.*


First, the result in Table 2 shows that the work stress faced by male managers has a negative influence on their career development behavior.

The value is −0.196, which is statistically significant (*p* < 0.001).

This result indicates that the when the stress faced by a male manager increases, it causes a decrease in their career development behavior.

Second, work stress experienced by female managers has a positive influence on their career development behavior.

The value is 0.473, which is statistically significant (*p* < 0.001).

This indicates that when the work stress experienced by female managers in the workplace increases, it leads to an increase in their career development behavior.

In conclusion, greater work stress decreases the career development behavior of male managers but strengthens the career development behavior of female managers.

**Hypothesis** **3** **(H3):**
*Work stress affects manager capabilities.*


First, as shown in the result in Table 1, work stress experienced by male managers has a negative influence on their capabilities.

The value is −0.190, which is statistically significant (*p* < 0.001).

This result shows the negative influence increased work stress on the capabilities of male managers.

Second, the work stress experienced by female managers has a positive influence on their capabilities.

The value is 0.145, which is statistically significant (*p* < 0.001) and demonstrates that increased work stress causes an increase in their capabilities.

Accordingly, a conclusion that greater work stress decreases the capabilities of male managers but strengthens the capabilities of female managers is suggested.

**Hypothesis** **4** **(H4):**
*Career development behavior plays a mediating role in the relationship between work stress and turnover intention.*


First, according to the results shown in Table 1 the mediation effect of the career development behaviors of male managers in relation to their work stress and turnover intention is not statistically significant.

The direct effect is −0.591, which is statistically significant (*p* < 0.05).

The indirect effect is −0.007, which is not statistically significant.

Second, the mediation effect of the career development behaviors of female managers in the relation to their work stress and turnover intention is statistically significant.

The direct effect is −0.117, which is statistically significant (*p* < 0.05).

The indirect effect is 0.094, which is statistically significant (*p* < 0.05).

Based on the results, it is suggested that career development behavior does not play the role of a mediator variable in the relationship between the work stress experienced by male managers and their turnover intentions but acts as a mediator variable in the relationship between the work stress experienced by female managers and their turnover intentions, and the effect strengthens their turnover intentions.

**Hypothesis** **5** **(H5):**
*Capabilities will play a mediating role in the relationship between work stress and turnover intention.*


As shown in Table 2, the mediation effect of the capabilities of male managers in the relation to their work stress and turnover intentions is statistically significant.

The direct effect is −0.591, which is statistically significant (*p* < 0.05).

The indirect effect is −0.038, which is statistically significant (*p* < 0.05).

Second, the mediation effect of the capabilities of female managers in the relation to their work stress and turnover intentions is statistically significant.

The direct effect is −0.117, which is statistically significant (*p* < 0.05).

The indirect effect is 0.100, which is statistically significant (*p* < 0.05).

This suggests that capabilities play a role as a mediator variable in the relationship between the work stress experienced by male managers and their turnover intentions and decreases their turnover intentions. In the case of female managers, capabilities play a role as a mediator variable in the relationship between their work stress and turnover intentions, and the effect strengthens their turnover intention.

## 5. Conclusions

According to the results of analyzing the influence of work stress of female and male managers in Korean companies on their turnover intentions, the following conclusions was reached:

First, the work stress experienced by managers increases their turnover intentions, regardless of gender.

Through this study, the effect of work stress on turnover intention was found to have an effect regardless of gender, suggesting that organizational efforts to reduce work stress are necessary. It seems that various policies and systems are needed in Korean companies to reduce work stress and increase job satisfaction. For example, job clarification [54,55] and work–life balance [56], which was proven to be effective in Korean companies in previous studies, should be developed more strongly at the organizational level to reduce work stress.

Second, greater work stress decreases the career development behaviors of male managers but strengthens the career development behaviors of female mangers. As a response to work stress, the responses of men and women are different in terms of career developing behaviors. From this result, we will need to discuss whether work stress is negative and why the responses according to gender are different. If career developing behavior is helpful as a facilitating factor for work stress reduction for female managers, it will be necessary to explore the appropriate degree of work stress. Therefore, it will also be necessary to measure work stress.

Third, greater work stress decreases the managerial capabilities of male mangers but strengthens the capabilities of female managers. The subjective evaluation of one’s own abilities differs according to gender. In other words, when work stress occurs, male managers evaluate their own abilities as low, and female managers evaluate them high. In other words, if female managers receive work stress, it will be necessary to consider whether it appears as a self-defense mechanism triggered by fear that their value will decrease as a human resource, or whether they are convinced that their value increases as they try harder to increase their value, that is, whether they are objectively aware of one’s own abilities.

Fourth, career development behavior does not play a role as a mediator variable in the relationship between the work stress experienced by male managers and their turnover intention but acts as a mediator variable in the relationship between the work stress experienced by female managers and their turnover intentions, and the effect strengthens their turnover intentions.

Fifth, capabilities play a role as a mediator variable in the relationship between the work stress experienced by male managers and their turnover intentions and decreases their turnover intentions. In the case of female managers, capabilities play a role as a mediator variable in the relationship between their work stress and turnover intentions, and the effect strengthens their turnover intentions.

Next, in the case of women, when the perception of career developing behaviors or their abilities increases, their turnover intention increases, and that of men decreases. It should be examined whether this affects the self-esteem or confidence of female managers because they do not have as many job offers or turnover opportunities as men do [53] as their career developing behaviors continue to increase in Korea, where they are socially underprivileged.

## 6. Discussion

Our study is to investigate the relation between the work stress perceived by managers, career development behaviors, capabilities, and turnover intentions as manager responses to stress at the workplace. It was found that the word stress experienced by managers increased their turnover intention regardless of gender. This result corresponded with the one obtained from the reference data [54,55]. It indicates that work stress is directly linked to turnover intention and correlates with the direct turnover rate of female and male managers.

Second, it was identified that when the work stress experience by male mangers increased, it caused a decrease in their career development behaviors, but increased work stress strengthened the career development behaviors of female mangers.

Male managers under stress become less motivated to do other tasks and became more passive in terms of career development activities such as participation in educational training, which is no different from previous research results [50,55]. The authors are more interested in the responses of female managers, to which work stress actually increases their career development behavior, as opposed to male managers. Based on the labor market context in Korea, where discriminatory characteristics against women in managerial positions are clear [13,14], these results were inferred as follows: The results can be interpreted as female managers considering stress as a crisis in their employment maintenance and a threat to their jobs and try to enhance their values as human resources. It should be noted that there is a difference in terms of the response to stress between the female and male managers in a male-centered organizational culture.

Third, it was identified that increased work stress decreased the capabilities of male mangers but strengthened the capabilities of female managers. A follow-up study is needed to determine whether a certain level of stress on female mangers is helpful for them to demonstrate their abilities, unlike male managers, and whether the female managers considered the demonstration of their abilities based on subjective awareness or on objective measurements. In this regard, it is necessary to determine how to manage work stress and how to help managers handle stress with professional support from qualified counselors in companies, with the intention of helping them achieve successful results and face positive career experiences rather than changing jobs. In addition, male and female managers should recognize that the impact of work stress is different, and there should be customized support according to gender.

Fourth, career development behavior did not act as a mediating variable in the relationship between the work stress experienced by male mangers and their turnover intentions but instead acted as a mediating variable in the relationship between when considering the work stress experienced by female mangers and their turnover intentions, and the effect strengthened the turnover intentions of the female mangers. This result was different from the one obtained from the reference data [46,48]. It was found that career development behavior worked differently depending on the female and male managers under stress. In the case of the male managers, career development behavior did not have a significant influence when they had turnover intentions due to increased stress [49,50]. Meanwhile, the turnover intentions of the female managers increased as their career development behavior increased. It can be inferred that they gained confidence in turnover after their values as human resources raised as a result of their career development behavior. A follow-up study should be undertaken to determine the causes of the differences in the responses to stress and its influence between female and male managers.

Fifth, capabilities acted as a mediating variable in the relation between the work stress experienced by male managers and their turnover intentions and decreased their turnover intention. In the case of the female managers, capabilities acted as a mediating variable in the relationship between their work stress and turnover intentions, and the effect was found to strengthen turnover intentions. It was found that management capabilities worked differently depending on the female and male managers under stress. In the case of the male managers, their turnover intentions due to increased stress highly influenced their capabilities [31,55]. When it came to the female managers, however, their capabilities were greatly influenced when they had turnover intentions because of increased stress. This shows that women have more difficulty in changing jobs or careers than men, as glass ceiling acts as a barrier when women rise to a managerial position in Korea’s organizational culture [57,58].

Gender issues within organizations have focused on inequality, causes of inequality, and intervention to relieve inequality. However, this study found that there are differences between men and women in their behaviors to cope with stress. Considering the differences between men and women rather than the differences between high and low values, gender-tailored human resource management and development from a perspective based not on relative deficiency but on difference are necessary.

The fact that there was a difference in terms of the types of leadership and ways of communication between men and women was discussed a lot in prior studies. This study found that there was a different way of responding to stress between men and women within organization. Follow-up studies on the analysis of the characteristics, categorization, and gaugeability of dealing with stress are required for the creation of mentally healthy organizational culture and the proper growth of female and male managers. Additionally, in this study, the relationship between work stress and turnover intention was examined based on previous research [59,60], and it was determined that turnover intentions affected the future turnover rate. However, since turnover intentions and actual b turnover behavior are clearly different, it is necessary to investigate the relation between work stress and turnover behavior and career development behavior as a moderating role through follow-up research. Since it is an additional research point, there is a difference in terms of the number of cases between men and women. Though this difference exists because it is based on panel data, the representativeness of the sample is secured.

Work stress causes disease, fraud, job dissatisfaction, or low organizational commitment, which increases turnover and can also negatively affect the organization’s work performance. Failure to control stress can disrupt work performance and reduce the potential for professional development. The work stress management of male and female managers is a factor that influences the sense of accomplishment in work and the outcome of organizational development. Therefore, programs for organizational emotional management and mental management will be needed.

## 7. Limitations

The limitations of this study are as follows: First, the questionnaire measuring the network variables was not sophisticated, which made it difficult to measure multiple questions grouped together. In other words, there was a limitation in terms of accurately measuring the responses to the questionnaires asking about work stress, career development capabilities, etc. Specifically, it did not reflect the respondents’ propensity to pursue both functions because they had to choose one of the two questionnaires measuring their turnover intentions. In the next survey, it will be necessary to elaborate on the variables related to the study according to the purpose of study.

Second, it is believed that the difference in the sample number between male managers and female managers is large, which may have affected the study results. Additionally, as described above, the male manager survey is not representative of all male managers in Korea, as fellow male managers of the same company in which female managers were extracted. Therefore, the results of this study have limitations in terms of generalization.

Third, a time series analysis was not performed due to the lack of data on the male managers. As the questionnaire on networks for the female and male mangers was limited to consistent data, it led to conducting a cross-sectional analysis. It is expected that more sophisticated analysis will be possible if a time series analysis is performed by supplementing the 2022 data.

## Figures and Tables

**Figure 1 ijerph-18-11119-f001:**
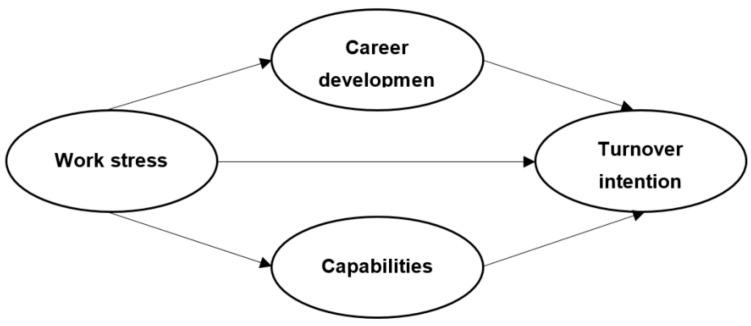
Research model.

**Table 1 ijerph-18-11119-t001:** Effect of work stress and career development behavior and capabilities on turnover intention.

Group	Independent Variable	Coeff	S.E.	t	*R* ^2^	F
Male	Constant	4.452	0.150	29.651 ***	0.414	232.499 ***
	Work stress	0.591	0.027	21.591 ***		
	Career development behavior	0.037	0.033	1.119		
	Capabilities	0.198	0.036	5.426 ***		
Female	Constant	0.897	0.056	16.036 ***	0.545	1002.874 ***
	Work stress	0.117	0.019	6.183 ***		
	Career development behavior	0.198	0.028	7.040 ***		
	Capabilities	0.688	0.028	24.472 ***		
Male	constant	3.724	0.082	45.685 ***	0.040	41.218 ***
	Work stress	−0.196	0.031	−6.420 ***		
Female	constant	1.768	0.053	33.539 ***	0.185	572.132 ***
	Work stress	0.473	0.020	23.919 ***		
Male	constant	2.404	0.107	22.553 ***	0.409	341.392 ***
	Work stress	−0.190	0.023	−8.190 ***		
	Career development behavior	0.535	0.024	22.672 ***		
Female	constant	0.583	0.038	15.377 ***	0.722	3263.918 ***
	Work stress	0.145	0.013	11.085 ***		
	Career development behavior	0.804	0.012	67.477 ***		

*** *p* < 0.001.

**Table 2 ijerph-18-11119-t002:** Result of mediation effect: capabilities (direct, indirect).

	Group	Direct Effect (LLCI, ULCI)	Indirect Effect (LLCI, ULCI)
Work stress → Turnover intention	Male	−0.591 (−0.645, −0.537)	−0.038 (−0.058, −0.019)
	Female	−0.117 (−0.154, −0.080)	0.100 (0.078, 0.122)

## Data Availability

Not applicable.

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
