# Peer review of "An Analysis of Male and Female Managers’ Responses to Work Stress: Focused on the Case of South Korea"

_ijerph, 2021, doi:10.3390/ijerph182111119_

Round 1

Reviewer 1 Report

Reviewer Title “ An analysis on the relation among work stress, career development behavior, capabilities, and turnover intention of female and male manageres: focuses on the case of south Korea.

Dear authors, the work presented here is of interest from the point of view of gender research. Showing the reality of heads of organizations in different countries can provide a vision of a reality that, despite the different regulations that have been approved at the international level, is still not brought to light, only the tip of the iceberg in the inequalities and inequalities that exist between women and men.

Despite the potential of this article, it is still in need of an important revision in order to have the necessary quality for its publication, and I am sure that you as authors will be able to do it.

The areas for improvement of this manuscript are listed point by point below.

Title:

The title is long and complicated, we should think of a shorter title that reflects that a genre analysis is going to be carried out. Authors should remember that it is not enough to say that a gender analysis is presented in men and women in the discussion.

Abstract:

Once the pertinent modifications have been made to the manuscript, the abstract should be redrafted according to the new version of the manuscript.

Introduction:

The introduction reflects an overall picture of the situation. How stress influences workers and team leaders and how inequalities between men and women are presented, despite this I think it is important to improve the discourse of gender inequalities and present the reality of the country where the study is circumscribed.

In this sense, for readers outside the country, we do not know how the territory is organized in terms of labor and the accessibility of men and women to the labor market. The type of work disaggregated by gender, for example, in Europe it is known that feminized work is more related to the social and health world. Likewise, how is gender mainstreaming in Korea, the glass ceiling, do women really reach the top positions?

In Europe there has also been an important social change with the incorporation of women into the labor market and with the change in the level of education, increasing those with high or medium education and decreasing those with low education. How is the reality in Korea?

Although it will be pointed out again in the scientific literature review section, but it is necessary to make a good previous literature review, there are very outdated citations and incorrectly placed citations, please review well all the literature cited (more information in the references section.

Methods:

Figure 1: Change the image to a higher quality one

3.2 Participants.

The authors state that they use as a source of information the Korean Women’s Manager Panel Survey 2018

More information about the survey is needed, as it is not clear how they select the participants and how it collects information from men, if it is a survey of women. This section should be expanded.

Lines 150-153:

This section explains the proportion of managers in companies of different sizes, since it is a gender study, it should be said how many managers there are in the different companies in order to show a picture of the reality of the situation. Do men and women really have the same proportion of representation in large and small companies? I do not know if it is possible to provide the following information, in this type of company is the Director of the company a man or a woman? I think it is necessary to show the top management positions by those who are represented.

Although I have seen that something is included in the limitations of the study, but a doubt arises, given the low representativeness of men, does this survey really show us the reality of the situation? I think this should be addressed and analyzed by the authors, if they really intend to make a study of gender inequality.

3.3 Data analysis ….

The variables included in the study are not explained, nor is how they are collected in the sample or anything else, this section mandatory in order to understand the study that has been carried out.

Has the Amos program actually been used and which model has been analyzed?

Tables 1, 2 and 3 could be organized into a single table.

On the other hand, the wordings of the results are repeated in the tables; it is advisable not to repeat this information. Examples are cited, lines 176-183, it is repetitive, perhaps with what is reflected in lines 185-187 expanding some information would be sufficient and the same for the part that is explained in Table 2 and Table 3.

Although it has been explained above, I do not know how the variables have been collected, it generates many doubts as to how stress has been measured and how it has been collected. Another question that arises is what is the rate of non-response and how has this fact been treated?

Tables 4 and 5, which show the results of the mediations, would perhaps be better to show the figures of how the variables are related rather than the tables with the results. Likewise, all the tables should be understood by themselves, therefore all the acronyms should be explained in the different table captions.

Another doubt that arises, have the analyses been adjusted for age? age is an important confounding variable and I do not believe that the reality of men and women will not be the same and even less so in the labor world, since the incorporation of women into the labor world has meant an important social change and a change in the role of care.

Although it will be repetitive, the analysis section should be improved and the adjustments and variables introduced in the different models should be explained in the different analyses carried out.

I repeat another comment, where are the models that have been made with AMOS? if I have not misunderstood, structural equations have been made, is that so? where are they?

  1. Conclusion

The conclusion should follow the discussion of the results, and this should be rewritten in depth and establish from the field of organizational health what measures can be taken into account to achieve effective equality between men and women. An approach to equality policies is necessary in this section, making an important analysis of the situation in Korea.

Discussion

The discussion should contain the following sections. Initially, a short summary (not repetitive of the results found by the authors) should be made.

Next, the results should be compared with other studies and other current research.

Continue with the limitations and strengths of the study to end with the conclusions and practical applications. Therefore, it is advised to restructure in this order.

The current discussion is all very confused and the studies used to compare results are outdated, studies from years such as 1993 (ref 48, 1992 ( ref 46) cannot be used.

It is necessary to carry out a good review of the literature and compare with the country's own studies, which in the current version is wrong and current.

Finally, the limitations should be reviewed since it is necessary to address the limitations of cross-sectional studies and survey studies and finally to include a good conclusion and practical applications of the study.

References

It has already been commented on different occasions, they are old and there are erroneous quotations, local studies are cited when they are European studies, obsolete quotations, from books……

Author Response

Thank you for your constructive feedback for our research. Please refer to the attached file for our response to the review.

Reviewer 2 Report

Research gap is described on page 1, line 81-3, methods on lines 129-30. The special focus is on gender differences in management, pages 4 following. The hypotheses on p. 2 summarize these ideas. Some repetition of the same very simple text (eg lines 191 f and lines 199 f) could be avoided

In future studies these hypotheses should be improved because "turnover intention" is clearly different to a turnover decision/action really taken. One could refer to "engagement" at the workplace which has been repeatedly measured by Gallup institute. Moreover, the theoretical COR background on page 2 could/should be enriched since the decisions by genders certainly depend on differing default options for male/female managers which certainly depend on the regulatory environment.

The data base used is of interest for the case of South Korea (even if it is qualified in the limitations at the end (which is also good).

Typos are in the figure on p. 4 (tureoever) and on line 143 (missing space before bracket).

Author Response

(The authors gave the same response as above.)

Round 2

Reviewer 1 Report

Dear authors, thank you for the important revision of your article, which has substantially improved its quality.

Here are some brief comments in case you would like to take them into account:

3.2. Participants

All the information provided is appreciated, but this paragraph requires bibliographic citations so that if the reader wants to check what has been said for other research, this is possible.

3.3. Data analysis

I am sorry because in the previous comments it was not clear but in the presentation of results you keep repeating data in tables and text. This is not correct and is redundant.

All the information provided is appreciated, but this paragraph requires bibliographic citations so that if the reader wants to check what has been said for other research, this is possible.

  1. Conclusion

I apologize for the repetition of my comment but it is that a conclusion should follow the discussion not the results. That comment was made previously and is still in the same place.

Moreover, the conclusion is limited to describing the main results without any implication from the point of view of gender and health analysis, gender and inequality. I believe that this part is the one that requires the most revision on the part of the authors.

  1. Discussion

I also recommend including a section on practical applications.

References

The bibliography has been slightly improved, but I believe that a greater effort should be made and the bibliographic citations should be better updated, and the DOI and URLs should be included.
